# Interconnection of Gut Microbiome and Efficacy of Immune Checkpoint Inhibitors in Inoperable Non-Small-Cell Lung Cancer

**DOI:** 10.3390/ijms26167758

**Published:** 2025-08-11

**Authors:** Fedor Moiseenko, Andrey Kechin, Maksim Koryukov, Ulyana Boyarskikh, Albina Gabina, Ani Oganesian, Sergey Belukhin, Maria Makarkina, Ekaterina Elsakova, Elizaveta Artemeva, Alexander Myslik, Nikita Volkov, Alexey Bogdanov, Ekaterina Kuligina, Svetlana Aleksakhina, Aglaya Iyevleva, Alexander Ivantsov, Andrey Bogdanov, Sergey Sidorenko, Vladimir Gostev, Alexey Komissarov, Vasilisa Dudurich, Lavrenty Danilov, Evgeny Imyanitov, Vladimir Moiseyenko

**Affiliations:** 1St. Petersburg Clinical Scientific and Practical Center for Specialized Types of Medical Care (Oncological) Named After N.P. Napalkov, Leningradskaya Str. 68A, Litera A, 197758 St. Petersburg, Russia; 2N.N. Petrov National Medical Research Center of Oncology, Ministry of Public Health of the Russian Federation, Leningradskaya Str. 68, 197758 St. Petersburg, Russia; 3Faculty of Oncology, North-Western State Medical University Named After I.I. Mechnikov, Kirochnaya Str. 41, 191015 St. Petersburg, Russia; 4Institute of Chemical Biology and Fundamental Medicine, Siberian Branch of the Russian Academy of Sciences, 8 Lavrentiev Avenue, 630090 Novosibirsk, Russia; 5Department of Medical Genetics and Oncology, Pediatric Oncology, and Radiation Therapy, Federal State Budgetary Educational Institution of Higher Education “Saint Petersburg State Pediatric Medical University” of the Ministry of Health of the Russian Federation, Litovskaya Str. 2, 194100 St. Petersburg, Russia; 6Federal State-Financed Institution Pediatric Research and Clinical Center for Infectious Diseases, the Federal Medical Biological Agency, Professor Popov Str. 9, 197022 St. Petersburg, Russia; 7Center of Genomic Technologies “Cerbalab”, Bolshoy Prospekt V.O. 90-2, 199106 St. Petersburg, Russia; vdudurich@cerbalab.ru (V.D.);

**Keywords:** microbiome, non-small cell lung cancer, immunotherapy, tumor microenvironment, immune checkpoint inhibitors, α-diversity, metagenome

## Abstract

The efficacy of immune checkpoint inhibitors (ICIs) in non-small-cell lung cancer (NSCLC) varies widely across patients. Growing evidence indicates that the gut microbiome, through its interaction with the tumor microenvironment, may influence the response to immunotherapy. To investigate this, we analyzed fecal and tumor samples from 63 patients with inoperable NSCLC undergoing ICI therapy. Based on microbiome profiling using 16S rRNA sequencing, patients were grouped according to treatment benefit, defined as progression-free survival (PFS) of six months or longer. Associations between α-diversity indices, microbial composition at the genus and phylum levels, and a composite Sum Index of Binary Abundance (SIBA) were examined in relation to clinical outcomes. Higher microbial α-diversity was linked to improved response to ICIs (*p*-value = 0.0078 for the Chao1 index). Multiple specific taxa, such as *Ruminococcus gauvreauii* (*p*-value = 2 × 10^−4^), *Ruminiclostridium* 9 (*p*-value = 8 × 10^−4^), and [*Eubacterium*] *ventriosum* (*p*-value = 9 × 10^−4^), were enriched in patients with favorable outcomes, whereas *Oscillibacter* and the *Eubacterium hallii* group were associated with disease progression (*p*-value = 2 × 10^−3^ and 9 × 10^−3^, respectively). The SIBA index, which reflects the absence of multiple beneficial bacterial taxa, proved to be a stronger predictor of treatment response than individual taxa alone. Median SIBA values were 18 vs. 24 in patients benefiting from IO therapy compared to non-responders (*p*-value = 9 × 10^−7^). These findings suggest that gut microbiome diversity and composition are closely tied to immunotherapy outcomes in NSCLC. Composite microbial metrics like SIBA may enhance predictive accuracy and inform personalized treatment approaches.

## 1. Introduction

Lung cancer, like many other malignancies, is increasingly recognized as a disorder stemming from a disrupted interaction between tumor cells and their surrounding tissue microenvironment [1]. Beyond intrinsic genetic aberrations, tumor progression is profoundly influenced by host factors, including immune surveillance and systemic physiological conditions [2]. Among these, the immune system plays a pivotal role, and recent therapeutic advances have harnessed this by developing immune checkpoint inhibitors (ICIs). These agents function by restoring anti-tumor immune activity, primarily through inhibition of the PD-1/PD-L1 axis, a key mechanism by which tumors evade immune detection [3].

The efficacy of ICIs hinges on two biological assumptions: first, that tumor cells are capable of eliciting an immune response; and second, that this response is actively suppressed by the tumor or its microenvironment via immune-inhibitory pathways. However, clinical outcomes with ICIs remain heterogeneous, suggesting the presence of additional, poorly understood modulators of therapeutic efficacy [4].

One emerging area of investigation is the gut microbiome, which plays a critical role in regulating systemic immunity [5]. Initial studies highlighted correlations between specific gut microbial compositions and ICI response. Landmark work by Routy and Zitvogel demonstrated that certain microbial profiles were associated with favorable responses to PD-1 blockade therapy, a finding that has since been replicated across diverse cancer types [6,7]. After identifying single-taxon correlations, next efforts were concentrated on the role of microbial profiles—specifically, the influence of combinations of bacterial species with both positive and negative effects not only on immunotherapy efficacy but also on the overall health of the meta-organism [8,9].

Subsequent research has shifted from descriptive correlations to mechanistic inquiries, probing how gut microbes influence immune homeostasis and tumor–immune dynamics [10,11,12,13,14]. Despite these efforts, consistent identification of specific bacterial taxa or pathways responsible for modulating ICI efficacy remains elusive. Variation across cohorts, tumor types, and geographies has further complicated efforts to standardize microbiome-based biomarkers [15].

In this study, we aimed to profile the gut microbiome of patients with inoperable non-small-cell lung cancer (NSCLC) treated with ICIs. By correlating microbiome diversity and composition with clinical outcomes, we sought to identify microbial signatures associated with therapeutic benefit, explore underlying mechanisms of immune modulation, and improve precision in patient selection for immunotherapy.

## 2. Results

### 2.1. Clinical Samples

This study included 63 patients with inoperable stages of NSCLC—IIIA, IIIB, IV—amendable for systemic treatment, who received immune oncology (IO) therapy between 2019 to 2021. Patients were treated with monotherapy IO regimens: 23 patients received atezolizumab or pembrolizumab as first-line therapy, while 40 patients were treated with atezolizumab or nivolumab as second-line therapy following platinum-based doublet chemotherapy) (Table 1).

Among the cohort, 1 patient achieved complete response, 12 achieved partial response, 31 had stable disease as the maximal effect of the treatment, and 15 experienced progressive disease. Median progression-free survival (PFS) for first-line therapy was 7 months (95% CI: 5.6–8.5), with a median overall survival (OS) of 29.5 months (95% CI: 25.8–33.3). For second-line therapy, median PFS was 6.3 months (95% CI: 4.7–7.9) and median OS was 18.5 months (95% CI: 8.2–28.8).

Irrespective of the line of therapy, we determined the benefit from IO therapy as the absence of disease progression within 6 months from the beginning of the treatment. Accordingly, clinical benefit was observed in 32 subjects, while lack of efficacy of IO drugs was noted in 31 subjects (Table 2).

We performed correlation analyses for all available clinical characteristics. No significant associations were found between treatment benefit and factors such as age, gender, neutrophil-to-lymphocyte ratio, or body mass index. However, PD-L1 expression in more than 50% of tumor cells and gender showed a statistically significant difference between cohorts with and without response to immune oncology therapy (*p*-value = 0.007 and 0.018, respectively).

### 2.2. Microbiome Analysis and Its Association with Clinical Characteristics

To investigate the association between microbiome indicators and clinical characteristics, we assessed three α-diversity metrics—Chao1, Shannon, and Simpson indices—which reflect microbial species diversity within individual samples. These metrics were evaluated across multiple rarefaction thresholds (Appendix A). For almost all thresholds, a significant difference in α-diversity was observed between patients who benefited from IO and those who did not (Table 3).

Chao1 median values were 207 versus 147 (*p*-value = 0.0078 in Mann–Whitney’s two-sided test), Shannon’s median values were 5.69 versus 5.44 (*p*-value = 0.041 in Mann–Whitney’s two-sided test), and Simpson’s median values were 0.958 versus 0.947 (*p*-value = 0.039). A significant decrease in Chao1 index value was also observed for patients of 72 years age and older versus others, at 136 versus 186 (*p*-value = 0.0285).

To assess whether the presence of specific bacterial taxa was associated with positive responses to immunotherapy or other clinical characteristics, we compared their abundance between patient groups using the two-sided Mann–Whitney test. Additionally, Fisher’s exact test was performed to evaluate whether taxa abundance could reliably classify patients into groups with favorable or unfavorable clinical outcomes (Appendix A).

At the phylum level, patients who benefited from immunotherapy exhibited a significantly higher abundance of *Bacillota* (0.488 vs. 0.419, Mann–Whitney’s *p*-value = 0.00057, Fisher’s exact test *p*-value = 0.024) and *Cyanobacteria* (0.0002 vs. 0, MW *p*-value = 0.061, FET *p*-value = 0.044), alongside a lower abundance of *Pseudomonadota* (0.0306 vs. 0.0692, MW *p*-value = 0.0088, FET *p*-value = 0.039) compared to patients without benefit. Similar patterns were observed for PFS, with higher *Firmicutes* abundance (threshold 0.403; log-rank *p* = 0.012) and lower *Pseudomonadota* abundance (threshold 0.079; log-rank *p* = 0.005) associated with improved outcomes. Regarding OS, only higher *Bacillota* abundance correlated significantly with longer survival (threshold 0.403; log-rank *p* = 0.030).

At the genus level, the taxa associated with clinical outcomes varied considerably. The most significant difference between patients with and without benefit from immunotherapy was observed for the [*Ruminococcus*] *gauvreauii* group, which showed a median abundance of 2.7 × 10^−4^ versus 0 in non-benefiting patients (Mann–Whitney’s *p* = 3.4 × 10^−4^, Fisher’s exact test *p* = 2.5 × 10^−4^). In contrast, *Coprobacter* was more frequently present in patients who experienced disease progression (median abundance 4.2 × 10^−4^ vs. 0; Mann–Whitney’s *p* = 1.8 × 10^−4^, Fisher’s exact test *p* = 3.3 × 10^−3^). For PFS, higher abundance of *Fusicatenibacter* correlated most strongly with improved outcomes (threshold 0.002; log-rank *p* = 9.8 × 10^−4^). Regarding OS, a higher abundance of the [*Eubacterium*] *hallii group* was associated with shorter survival (threshold 0.002; log-rank *p* = 5.9 × 10^−4^). These genus-level taxa with differential abundance across clinical groups are summarized in Figure 1. In addition to PFS ≥ 6 months after immunotherapy (benefit_6 in Figure 1), we also included PFS ≥ 10 months (benefit_10) in the analysis.

The lists of taxa differing between patient groups across clinical characteristics varied considerably. This variability suggests that the observed associations may be driven more by the presence of specific metabolic pathway genes distributed across different organisms rather than by any single bacterial species. Accordingly, favorable clinical outcomes or responses to therapy may depend not on a single pathway or species but on a broader set of functional pathways collectively contributed by multiple members of the microbiome. Based on this concept, we aimed to identify a set of bacterial taxa whose presence or absence—and insufficient abundance—simultaneously influence multiple clinical characteristics (Table 4). Abundance thresholds were independently determined for each bacterial species based on values yielding the lowest *p*-value in Fisher’s exact test. Thresholds were higher for consistently abundant bacteria (e.g., Faecalibacterium) and lower for bacteria frequently undetected in subsets of samples (e.g., Holdemania). We propose that while common bacterial taxa may provide general metabolic pathways (e.g., vitamin biosynthesis) to the host organism, rare taxa could contribute enzymes for digesting or synthesizing specialized compounds. In this analysis, we omitted regression analyses and machine learning approaches as they require validation using an independent sample cohort.

Using the thresholds of presence defined for each genus and phylum, the sum index of binary abundance (SIBA) was calculated for all patients as follows:$$SIBA=\sum_{X}{b(x}_{i},t_{i})-\sum_{Y}{b(y}_{j},t_{j}),where\;b\left(x,t\right)=\left\{\begin{array}{c} 1,\;\mathrm{if}\;x\geq t,\\ 0,\;\mathrm{else}; \end{array}\right.,$$

X and Y represent bacterial taxa included in SIBA with lower and higher abundance in patients with advantageous or disadvantageous clinical characteristics, respectively; x_i_ and y_i_ are abundance values for corresponding taxa; and t_i_ is the threshold for taxon abundance to be considered as present in the microbiome. Therefore, a lower SIBA value indicates a group with a higher probability of response to IO therapy.

We then analyzed the correlation between SIBA and all clinical characteristics (Table 5), finding the strongest associations with differential benefit from IO therapy (*p* = 6 × 10^−7^ by Mann–Whitney’ test; *p* = 9 × 10^−7^ by Fisher’s exact test), PFS (*p* = 6 × 10^−9^ by log-rank test), and OS (*p* = 1 × 10^−2^ by log-rank test). Kaplan–Meier survival curves are shown in Appendix A. The SIBA index successfully stratified patients into groups with distinct PFS, OS, and IO therapy benefit outcomes, while no single bacterial taxon demonstrated similarly robust associations (Figure 2).

## 3. Discussion

In this study, we report the correlation between various microbiome-related factors and clinical outcomes in patients with NSCLC treated with ICIs.

Our data confirm the association between microbiome diversity, particularly α-diversity indices such as Chao1, Shannon, and Simpson, and clinical benefit from IO treatment. Patients with clinical benefit (defined as absence of disease progression for ≥6 months) consistently exhibited higher microbial diversity, reflected in elevated Chao1 (median 207 vs. 147; *p*-value = 0.0078), Shannon, and Simpson indices. These findings align with prior studies demonstrating that gut microbiome diversity positively correlates with enhanced immune responses and improved immunotherapy efficacy [16]. However, while microbiome diversity showed a significant association with treatment outcomes in our cohort, it did not fully explain response variability, highlighting the need for more comprehensive analyses of microbiome–host interactions in cancer treatment.

Multiple previous studies have identified specific fecal microbiota—particularly *Akkermansia muciniphila* and *Bifidobacterium*, the most extensively studied in lung cancer patients—as being associated with durable clinical benefit from ICI therapy [14,16,17,18]. In our analysis, neither of these species showed significant association with treatment efficacy. The probable explanation might be their lower concentration due to age-related *Bifidobacteria* concentration, *Bacteroides* dominance, and regional pattern, as all the patients originated from northwestern region of Russia [19,20]. Instead, we identified several other key genera demonstrating treatment-related associations. Some of these, such as members of the *Ruminococcaceae* family, have previously been linked to a healthy microbiome and less aggressive tumors [21]. Notably, the **Ruminococcus gauvreauii** group exhibited significantly higher abundance in patients achieving clinical benefit from IO therapy (*p*-value = 2 × 10^−4^), whereas *Coprobacter* was more prevalent in those with disease progression (*p*-value = 3 × 10^−3^). Additionally, *Lachnospiracea* (*p* = 0.02), *Bacteroidetes* (*p* = 0.0496), * Faecalibacterium* (*p* = 0.01), * Dorea* (*p* = 0.01), and * Firmicutes* (*p* = 0.02)—enriched in out cohort among patients benefiting from checkpoint inhibitors—have also been previously reported as favorable [22,23,24]. In contrast, other taxa, including Oscillibacter* and * Holdemania* (previously associated with positive outcomes [22]) were instead correlated with treatment insensitivity in our study (enriched in patients with disease progression, *p* = 0.002 and *p* = 0.001, respectively). These findings underscore the importance of a profile-based approach rather than focusing on individual taxa.

The microbiota associated with immunotherapy benefit may potentiate anti-tumor immunity by modulating favorable immune response [18]. Conversely, a less favorable microbiome composition could promote immunosuppressive mechanisms or tumor-supporting microbial functions. Such effects may involve secreted antigens or metabolites that either exhaust immune T cells, activate them, or modulate their activity [25]. At the same time, other bacteria with specific metabolic pathways may break down harmful metabolites or antigens, thereby enhancing the efficacy of subsequent immunotherapy.

Several consecutive studies summarizing research efforts in this field confirmed a general association between the microbiome and treatment response but revealed limited reproducibility of specific microbial species as reliable biomarkers [15]. The microbial species identified as influential in our cohort showed limited relevance in other patient populations. These discrepancies may be attributed to geographic variations and local dietary habits among the studied groups. To better capture the complexity of microbiome-mediated effects, we developed SIBA, a composite metric integrating the presence of multiple microbial taxa. SIBA demonstrated superior predictive power for treatment response compared to individual taxa analysis and outperformed conventional methods in stratifying patients by clinical outcomes. This integrative approach offers clinicians a more practical and robust tool, as it reflects the collective influence of the microbiome on immunotherapy efficacy, rather than relying on single taxa, which often exhibit inconsistent associations with treatment response [7].

Our study has several limitations. Although SIBA demonstrated promising predictive performance, the retrospective design and relatively small sample size constrain the generalizability of the findings. Independent cohort validation remains necessary to confirm these results. Future studies should also address the lack of *p*-value adjustment for multiple comparisons, which was not feasible in the current investigation due to limited sample size. This limitation could also explain the only numerical improvement in PFS and OS for patients with high PD-L1 expression (above 50%) compared to other patients (PFS in PD-L1 low vs. high: 6.3 (4.2–8.4) vs. 16.3 (2.4–30.2); *p* = 0.161). A borderline numerically significant difference in survival between male and female patients was also observed (18.5 (7.7–29.3) vs. 40.2 (28.4–52.0); *p* = 0.087).

An additional limitation is the small number of clinical variables included in the analysis. For example, we were unable to collect data and analyze the role of prior antibiotic use before immunotherapy. Moreover, although we identified associations between microbiome features and clinical outcomes, the underlying causal mechanisms remain unknown. Future research should prioritize larger, multicenter cohorts with longitudinal sampling to validate these associations and elucidate the biological pathways through which the microbiome influences immunotherapy efficacy. Integrating microbiome profiles with tumor genomic data and immune landscape characterization would provide a more comprehensive understanding of how microbial diversity interacts with the tumor microenvironment and modulates antitumor immune responses.

## 4. Materials and Methods

### 4.1. Clinical Samples

Fecal samples were collected between 2019 and 2021 from patients diagnosed with NSCLC who were eligible to receive ICIs at any line of therapy. Eligibility criteria included an inoperable disease status, indication for immunotherapy based on current local clinical guidelines and thoracic multidisciplinary team (MDT) consensus, and provision of informed consent for participation in microbiome and tumor microenvironment molecular analyses.

Treatment decisions were made through MDT discussions. No antibiotics were used in the treatment. Clinical outcomes were assessed based on RECIST v1.1 criteria. PFS was defined as the time from the first dose of immunotherapy to the date of radiologically confirmed disease progression, death, or last follow-up, whichever occurred first. OS was measured from the initiation of immunotherapy to the date of death or last follow-up.

Fecal samples for microbiome analysis were collected prior to the first immunotherapy dose. Samples were transferred into 1.5 mL Eppendorf tubes containing a mucolytic transport medium and stored at −20 °C until further processing.

### 4.2. 16S rRNA Gene Sequencing

Total DNA was isolated from fecal specimens following bead-beating homogenization in lysis buffer (0.1 M Tris–HCl, 0.1 M EDTA, 0.01 M NaCl, 0.5% sodium dodecyl sulfate, pH 8.0) with subsequent DNA extraction using a DNeasy Blood & Tissue Kit (Qiagen, Hilman, Germany), following the manufacturer’s protocol. The V3–V4 hypervariable regions of the 16S rRNA gene were amplified using primers previously described by Klindworth et al. [20], in combination with Phusion Hot Start II High-Fidelity PCR Master Mix (Thermo Scientific, Waltham, MA, USA). Each PCR reaction contained 5 ng of template DNA and was subjected to the following thermal cycling conditions: initial denaturation at 94 °C for 3 min; 35 cycles of denaturation at 94 °C for 30 s, annealing at 55 °C for 30 s, and extension at 72 °C for 30 s; followed by a final extension at 72 °C for 5 min.

Sequencing of the amplified 16S rRNA gene region was performed on the Illumina MiSeq platform using a MiSeq Reagent Kit v3 (600-cycle, Illumina, San Diego, CA, USA), according to the manufacturer’s recommendations.

### 4.3. Reads Processing

Sequence reads were processed using QIIME2 [21] with the following workflow. Raw NGS reads were preprocessed with the DADA2 [22] “denoise-paired” command followed by detecting and filtering chimeras with the vsearch [23] “uchime-denovo” command. The sequences left after the filtration were clustered with the vsearch “cluster-features-de-novo” command, and taxonomy was assigned for each cluster with the “classify-consensus-blast” command of the “feature-classifier” QIIME 2 tool and SILVA version 132 database [24]. Taxonomic assignments with confidence thresholds below 0.97 were excluded to ensure reliability. The abundance value was evaluated as a ratio of operational taxonomic unit (OTU) counts to the total number of reads for the sample left after all preprocessing steps.

### 4.4. Statistical Analysis of α-Diversity

The α-diversity Shannon, Simpson, and Chao1 indices were evaluated in 20 iterations of OTU tables rarefied to 5000 sequence reads per sample using the QIIME2 diversity module. The rarefaction threshold was determined based on rarefaction curve saturation: 5000 sequences per sample were sufficient to reach plateau conditions for diversity indices while maximizing the number of samples retained in the analysis. The Kaplan–Meier log-rank test was carried out with the lifelines python package [25] and alpha-value = 0.99. As examples, Kaplan–Meier survival curves for patients with high and low SIBA values are shown in Appendix A.

## 5. Conclusions

The current study reinforces the significant association between gut microbiome diversity and clinical benefit from ICIs in patients with NSCLC. While α-diversity metrics correlated with improved treatment outcomes, specific microbial species demonstrated variable relevance across populations, underscoring the complexity of microbiome–host interactions. The development of the SIBA composite metric highlights the potential of integrative approaches to improve predictive accuracy beyond single-taxon analyses. Despite the limitations of our retrospective design and sample size, our findings contribute to a growing body of evidence supporting the microbiome’s role in modulating immunotherapy efficacy. Future prospective, large-scale, multi-omics studies are crucial to elucidate the mechanisms driving differential responses to immune checkpoint inhibitors and to facilitate the integration of microbiome-based biomarkers into clinical practice for improved therapeutic outcomes.

## Figures and Tables

**Figure 1 ijms-26-07758-f001:**
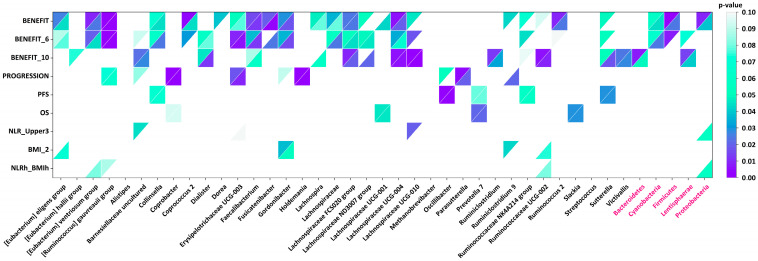
Association of clinical characteristics with abundance of genus and phylum taxa. A higher number of distinct taxa were significantly associated with benefit from IO therapy than with other characteristics. Mann–Whitney’s (upper left triangles) and Fisher’s exact test (lower right triangles) *p*-values for different genus (black names) and phylum (pink names) taxa (horizontal axis), in which abundance values significantly differed between patient groups subdivided by clinical characteristics (vertical axis). Benefit_6—PFS ≥ 6; Benefit_10—PFS ≥ 10; PFS—progression-free survival; OS—overall survival; NLR—ratio of neutrophil to lymphocyte numbers; BMI—body mass index; NLRh_BMIh—high value of NLR (more than T3 of the population studied) and high value of BMI (≥29).

**Figure 2 ijms-26-07758-f002:**
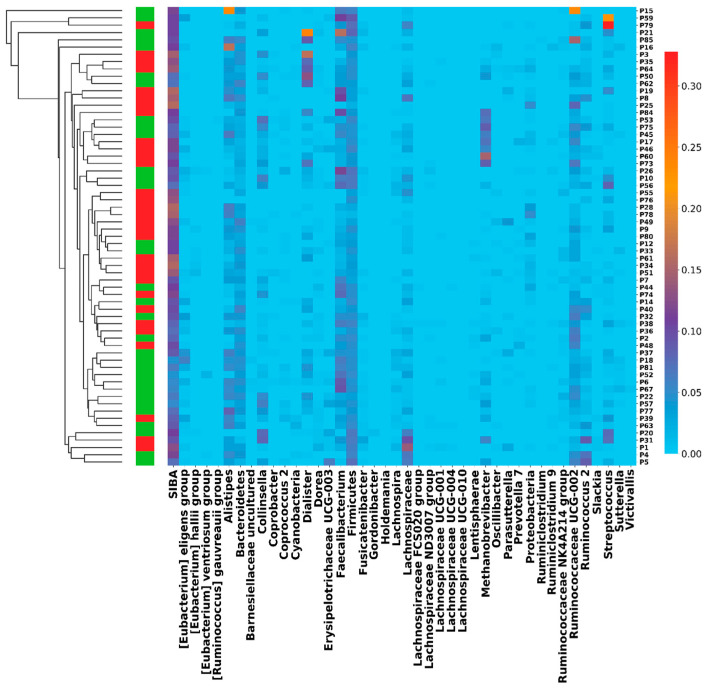
Hierarchical clustering of abundance values for genus and phylum taxa for patients with benefit (green left blocks) or without benefit (red left blocks) from IO. The abundance values for phylum taxa were divided by 10 to normalize with other values. SIBA values were divided by 200.

**Table 1 ijms-26-07758-t001:** Patient treatment regimens, response outcomes, and reported toxicities.

		Benefit					
		No		Yes		Total	
		N	%	N	%	N	%
**Line of ICIs**	**First-line**	10	32.3%	13	40.6%	23	36.5%
	Pembrolizumab	6	60%	9	69.2%	15	
	Atezolizumab	4	40%	4	30.8%	8	
	**Second-line**	21	67.7%	19	59.4%	40	63.5%
	Pembrolizumab	16	76.2%	12	63.2%	28	
	Atezolizumab	4	19.0%	2	10.5%	6	
	Nivolumab Ipilimumab	1	4.8%	5	26.3%	6	
**Toxicity**	No	24	88.9%	26	81.3%	50	84.7%
	Liver	0	0.0%	2	6.3%	2	3.4%
	Skin	0	0.0%	1	3.1%	1	1.7%
	Thyroid	0	0.0%	2	6.3%	2	3.4%
**Response**	N/A	2	6.5%	1	3.2%	3	4.8%
	Complete response	0	0.0%	1	3.2%	1	1.6%
	Partial response	3	9.7%	9	29.0%	12	19.4%
	Stable	12	38.7%	19	61.3%	31	50.0%
	Disease	14	45.2%	1	3.2%	15	24.2%

**Table 2 ijms-26-07758-t002:** Clinical characteristics of patients included in the microbiome analysis. BMI—body mass index; NLR—ratio of neutrophil to lymphocyte number; T3—upper tertile of the whole cohort values.

	Benefit						
	No		Yes		Total		
Characteristics	N	%	N	%	N	%	*p*-Value
**Gender**							0.018
Male	28	90.3%	21	65.6%	49	77.8%	
Female	3	9.7%	11	34.4%	14	22.2%	
**Age**							0.916
<65	13	41.9%	13	40.6%	26	41.3%	
≥65	18	58.1%	19	59.4%	37	58.7%	
**BMI ≥ 29**							0.591
No	18	58.1%	16	50.0%	34	54.0%	
Yes	11	35.5%	15	46.9%	26	41.3%	
NA	2	6.5%	1	3.1%	3	4.8%	
**NLR ≥ T3**							0.124
No	24	77.4%	19	59.4%	43	68.3%	
Yes	7	22.6%	13	40.6%	20	31.7%	
**NLR ≥ T3 and BMI ≥ 29**							0.758
No	27	87.1%	27	84.4%	54	85.7%	
Yes	4	12.9%	5	15.6%	9	14.3%	
**Histology**							0.259
Adenocarcinoma	14	45.2%	19	59.4%	33	52.4%	
Squamous cell	17	54.8%	13	40.6%	30	47.6%	
**Activating mutations**							0.721
None	28	90.3%	29	90.6%	57	90.5%	
EGFR	1	3.2%	1	3.1%	2	3.2%	
ALK/ROS1	1	3.2%	0	0.0%	1	1.6%	
KRAS	1	3.2%	2	6.3%	3	4.8%	
**PD-L1 staining**							0.007
<50%	27	87.1%	18	56.3%	45	71.4%	
>50%	4	12.9%	14	43.8%	18	28.6%	

**Table 3 ijms-26-07758-t003:** Statistically significant differences in α-diversity indices for patient groups subdivided by values of clinical characteristics. For all indices, median values for two groups and *p*-values are provided. Coverage threshold values in the table are rounded.

Clinical Characteristic	Coverage Depth Threshold	Chao1	Shannon	Simpson
Benefit	1000–8000	207 vs. 147 (0.0078)	5.69 vs. 5.44 (0.041)	0.958 vs. 0.947 (0.039)
Age ≥ 72	3000–7000	136 vs. 186 (0.0285)	5.44 vs. 5.57 (0.232)	0.953 vs. 0.955 (0.282)

**Table 4 ijms-26-07758-t004:** Bacterial taxa clinically significantly associated with clinical characteristics and included into the SIBA index with the thresholds provided.

Taxon	Bacterial Genus/Group/Phylum	Associated with	Abundance Threshold
Genus	[*Eubacterium*] *eligens group*	Benefit	0.01
Genus	[*Eubacterium*] *hallii group*	Progression	0.002
Genus	[*Eubacterium*] *ventriosum group*	Benefit	0.0001
Genus	[*Ruminococcus*] *gauvreauii group*	Benefit	0.0001
Genus	*Alistipes*	Benefit	0.05
Phylum	Bacteroidetes	Benefit	0.36
Genus	*Uncultured genus from Barnesiellaceae family*	Benefit	0.00007
Genus	*Collinsella*	Benefit	0.009
Genus	*Coprobacter*	Progression	0.0002
Genus	*Coprococcus 2*	Benefit	0.0001
Phylum	Cyanobacteria	Benefit	0.0001
Genus	*Dialister*	Benefit	0.01
Genus	*Dorea*	Benefit	0.0001
Genus	*Erysipelotrichaceae UCG-003*	Progression	0.0008
Genus	*Faecalibacterium*	Benefit	0.03
Phylum	Firmicutes	Benefit	0.4
Genus	*Fusicatenibacter*	Benefit	0.002
Genus	*Gordonibacter*	Benefit	0.01
Genus	*Holdemania*	Progression	0.00006
Genus	*Lachnospira*	Benefit	0.0001
Genus	*Lachnospiraceae*	Benefit	0.006
Genus	*Lachnospiraceae FCS020 group*	Benefit	0.0001
Genus	*Lachnospiraceae ND3007 group*	Benefit	0.001
Genus	*Lachnospiraceae UCG-001*	Benefit	0.001
Genus	*Lachnospiraceae UCG-004*	Benefit	0.0001
Genus	*Lachnospiraceae UCG-010*	Benefit	0.00004
Phylum	Lentisphaerae	Benefit	0.0003
Genus	*Methanobrevibacter*	Benefit	0.005
Genus	*Oscillibacter*	Progression	0.002
Genus	*Parasutterella*	Progression	0.0001
Genus	*Prevotella 7*	Progression	0.001
Phylum	Proteobacteria	Progression	0.05
Genus	*Ruminiclostridium*	Benefit	0.00004
Genus	*Ruminiclostridium 9*	Benefit	0.0001
Genus	*Ruminococcaceae NK4A214 group*	Benefit	0.002
Genus	*Ruminococcaceae UCG-002*	Benefit	0.01
Genus	*Ruminococcus 2*	Benefit	0.01
Genus	*Slackia*	Progression	0.002
Genus	*Streptococcus*	Progression	0.006
Genus	*Sutterella*	Benefit	0.002
Genus	*Victivallis*	Benefit	0.00004

**Table 5 ijms-26-07758-t005:** Association of SIBA index value with clinical characteristics: progression, benefit from IO therapy, body mass index (BMI), neutrophil-to-lymphocyte numbers ratio (NLR), simultaneously high value of BMI and NLR (BMIh_NLRh), progression-free survival (PFS), and overall survival (OS) values. For the NLR, the upper tertile value (T3) was used for patient stratification. Mann-Whitney’s two-sided test was applied. For PFS and OS analysis, log-rank test *p*-values are provided.

Characteristics	Progression	Benefit	BMI ≥ 29	NLR > T3	BMIh_NLRh	PFS	OS
Median SIBA (Yes)	21.0	18.0	19.5	19.5	19.0		
Median SIBA (No)	19.5	24.0	21.0	21.0	21.0		
Mann–Whitney’s *p*-value	0.70	**6 × 10^−7^**	0.87	0.40	0.84		
Threshold for the Fisher’s exact test/log-rank test	20.25	21	20.25	20.25	20	23	23
Fisher’ exact test/log-rank test *p*-value	0.31	**9 × 10^−7^**	0.61	0.29	0.48	**6 × 10^−9^**	**0.01**

## Data Availability

The raw data supporting the conclusions of this article will be made available by the authors on request.

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
