# Peer review of "Interconnection of Gut Microbiome and Efficacy of Immune Checkpoint Inhibitors in Inoperable Non-Small-Cell Lung Cancer"

_ijms, 2025, doi:10.3390/ijms26167758_

Round 1
Reviewer 1 Report
Comments and Suggestions for Authors
The manuscript "Interconnection of Gut Microbiome and Efficacy of Immune Checkpoint Inhibitors in Inoperable Non-Small Cell Lung Cancer" comprehensively explores the association between gut microbiome diversity, specific microbial taxa, and clinical outcomes in patients with inoperable non-small cell lung cancer undergoing immune checkpoint inhibitor therapy. Utilizing 16S rRNA gene sequencing of fecal samples, the study introduces a composite microbial biomarker, the Sum Index of Binary Abundance (SIBA), which aims to enhance the prediction of treatment responses. This research addresses a pertinent and clinically significant question within the field of immuno-oncology, contributing to the expanding body of evidence that connects gut microbiota to immunotherapy outcomes. The cohort is well characterized, the methodology is clearly articulated, and the integration of microbial and clinical data is commendable.
I acknowledge the authors for their diligent efforts in designing and conducting this multifaceted study. The development of the SIBA index represents an innovative approach to addressing the limitations associated with single-taxon associations, providing a pathway toward more reproducible microbiome-based predictors. However, several aspects of the manuscript would benefit from additional clarification and discussion to enhance interpretability, transparency, and overall impact.
Below are a few suggestions that would strengthen the manuscript.
Major comments:
- Both PD-L1 expression and sex are significantly associated with clinical benefit in this cohort, as indicated in Table 2. However, the manuscript does not explore their potential influence on the reported microbiome-outcome associations. While the study establishes links between microbial features and clinical benefit, it does not assess whether these associations are independent of known predictors of clinical benefit. Including a discussion of this limitation would enhance transparency and provide a more nuanced context for the findings.
- The rationale for developing the SIBA index is thoughtfully presented, with clear explanations of its conceptual structure. This approach is well justified as a response to the reproducibility challenges posed by single-taxon markers. However, the manuscript would benefit from additional details regarding how specific genus- and phylum-level taxa were selected for inclusion, as well as how the abundance thresholds were determined. Clarifying whether the selection was based on statistical significance, consistency across outcomes, or observed optimization would strengthen reproducibility. Moreover, while SIBA demonstrates promising predictive performance within this cohort, the absence of internal validation limits its generalizability. A brief discussion of this limitation would enhance the methodological rigor of the study.
- The manuscript includes a wide range of statistical comparisons across α-diversity metrics, taxonomic features, and clinical outcomes. However, it does not address the need for correction for multiple hypothesis testing. This consideration is particularly relevant given the number of comparisons and the presence of several borderline p-values (such as p-values for Chao1, Shannon, and Simpson). While exploratory microbiome studies often report unadjusted p-values, acknowledging this limitation would improve transparency and facilitate appropriate interpretation of the findings.
- While the manuscript presents numerous microbiome-clinical associations supported by appropriate statistical methods, it lacks a structured discussion section to provide an in-depth interpretation of these findings. A more comprehensive understanding of the results, however concise, would better connect the data to the biological context, highlight novelty or consistency with existing literature, and enhance the clarity and depth of the manuscript. In addition to a more structured interpretation of their findings, integrating relevant literature on gut microbiome interactions specifically in NSCLC, including similarities, discrepancies, or gaps compared to recent studies, would help contextualize the results and highlight their contribution to the field.
Minor comment:
The use of 5,000 reads per sample for rarefaction is both appropriate and common in α-diversity analyses. Nevertheless, providing a brief rationale, such as referencing rarefaction curve saturation, sequencing coverage, or alignment with field norms, would improve transparency and assist readers unfamiliar with this method in understanding the basis for this choice.
In summary, this study presents an innovative and well-organized investigation of an essential question in the field of microbiome and cancer immunotherapy research. The integration of clinical, microbial, and statistical frameworks constitutes a significant strength, and the SIBA index represents a novel contribution with potential for future development. Addressing the comments outlined above, particularly those related to confounders, validation, and interpretive depth, will enhance the robustness and clarity of the manuscript. I commend the authors and express my gratitude for their contribution.
Author Response
1. Both PD-L1 expression and sex are significantly associated with clinical benefit in this cohort, as indicated in Table 2. However, the manuscript does not explore their potential influence on the reported microbiome-outcome associations. While the study establishes links between microbial features and clinical benefit, it does not assess whether these associations are independent of known predictors of clinical benefit. Including a discussion of this limitation would enhance transparency and provide a more nuanced context for the findings.
Response 1: In accordance with Referee 1’s comment, we have added discussion of this limitation in the corresponding section (lines 262–271):
Our study has several limitations. Although SIBA demonstrated promising predictive performance, the retrospective design and relatively small sample size constrain the generalizability of the findings. Independent cohort validation remains necessary to confirm these results. Future studies should also address the lack of p-value adjustment for multiple comparisons, which was not feasible in the current investigation due to limited sample size. This limitation could also explain the only numerical improvement in PFS and OS for patients with high PD-L1 expression (above 50%) compared to other patients (PFS in PD-L1 low vs. high: 6.3 (4.2-8.4) vs 16.3 (2.4-30.2); Ñ€ = 0.161). A borderline numerically significant difference in survival between male and female patients was also observed (18.5 (7.7-29.3) vs 40.2 (28.4 – 52.0); p = 0.087).
2. The rationale for developing the SIBA index is thoughtfully presented, with clear explanations of its conceptual structure. This approach is well justified as a response to the reproducibility challenges posed by single-taxon markers. However, the manuscript would benefit from additional details regarding how specific genus- and phylum-level taxa were selected for inclusion, as well as how the abundance thresholds were determined. Clarifying whether the selection was based on statistical significance, consistency across outcomes, or observed optimization would strengthen reproducibility. Moreover, while SIBA demonstrates promising predictive performance within this cohort, it limits its generalizability. A brief discussion of this limitation would enhance the methodological rigor of the study.
Response 2: In line with suggestion, we have added the description of SIBA construction process in more details (lines 172–182):
Based on this concept, we aimed to identify a set of bacterial taxa whose presence or absence—and insufficient abundance—simultaneously influence multiple clinical characteristics (Table 4). Abundance thresholds were independently determined for each bacterial species based on values yielding the lowest p-value in Fisher's exact test. Thresholds were higher for consistently abundant bacteria (e.g., Faecalibacterium) and lower for bacteria frequently undetected in subsets of samples (e.g., Holdemania). We propose that while common bacterial taxa may provide general metabolic pathways (e.g., vitamin biosynthesis) to the host organism, rare taxa could contribute enzymes for digesting or synthesizing specialized compounds. In this analysis, we omitted regression analyses and machine learning approaches as they require validation using an independent sample cohort.
The discussion of the limits raised by the absence of internal validation has also been added (lines 246–247).
3. The manuscript includes a wide range of statistical comparisons across α-diversity metrics, taxonomic features, and clinical outcomes. However, it does not address the need for correction for multiple hypothesis testing. This consideration is particularly relevant given the number of comparisons and the presence of several borderline p-values (such as p-values for Chao1, Shannon, and Simpson). While exploratory microbiome studies often report unadjusted p-values, acknowledging this limitation would improve transparency and facilitate appropriate interpretation of the findings.
Response 3: In accordance with the suggestion of the Review 1, we have added the discussion of this limitation (lines 265–267).
Future studies should also address the lack of p-value adjustment for multiple comparisons, which was not feasible in the current investigation due to limited sample size.
4. While the manuscript presents numerous microbiome-clinical associations supported by appropriate statistical methods, it lacks a structured discussion section to provide an in-depth interpretation of these findings. A more comprehensive understanding of the results, however concise, would better connect the data to the biological context, highlight novelty or consistency with existing literature, and enhance the clarity and depth of the manuscript. In addition to a more structured interpretation of their findings, integrating relevant literature on gut microbiome interactions specifically in NSCLC, including similarities, discrepancies, or gaps compared to recent studies, would help contextualize the results and highlight their contribution to the field.
Response 4: We thank Reviewer 1 for raising this important point. In response to this comment, we have substantially improved the discussion of the study findings (lines 222–248):
Multiple previous studies have identified specific fecal microbiota – particularly Akkermansia muciniphila and Bifidobacterium, the most extensively studied in lung cancer patients – as being associated with durable clinical benefit from ICI therapy [12,14–16]. In our analysis, neither of these species showed significant association with treatment efficacy. Instead, we identified several other key genera demonstrating treatment-related associations. Some of these, such as members of the Ruminococcaceae family, have previously been linked to a healthy microbiome and less aggressive tumors [17]. Notably, *Ruminococcus gauvreauii* group exhibited significantly higher abundance in patients achieving clinical benefit from IO therapy (p-value = 2*10-4), whereas Coprobacter was more prevalent in those with disease progression (p-value = 3*10-3). Additionally, *Lachnospiracea* (p=0.02), *Bacteroidetes* (p=0.0496), * Faecalibacterium* (p=0.01), * Dorea* (p=0.01), and * Firmicutes* (p = 0.02) – enriched in out cohort among patients benefiting from checkpoint inhibitors – have also been previously reported as favorable [18–20]. In contrast, other taxa, including *Oscillibacter* and *Holdemania* (previously associated with positive outcomes [18]) were instead correlated with treatment insensitivity in our study (enriched in patients with disease progression, p = 0.002 and p = 0.001, respectively). These findings underscore the importance of a profile-based approach rather than focusing on individual taxa.
The microbiota associated with immunotherapy benefit may potentiate anti-tumor immunity by modulating favorable immune response [16]. Conversely, a less favorable microbiome composition could promote immunosuppressive mechanisms or tumor-supporting microbial functions. Such effects may involve secreted antigens or metabolites that either exhaust immune T cells, activate them, or modulate their activity [21]. At the same time, other bacteria with specific metabolic pathways may break down harmful metabolites or antigens, thereby enhancing the efficacy of subsequent immunotherapy.
5. The use of 5,000 reads per sample for rarefaction is both appropriate and common in α-diversity analyses. Nevertheless, providing a brief rationale, such as referencing rarefaction curve saturation, sequencing coverage, or alignment with field norms, would improve transparency and assist readers unfamiliar with this method in understanding the basis for this choice.
Response 5: As it was noticed by the Reviewer, we have added the following rationale for the chosen threshold (lines 326–328):
The rarefaction threshold was determined based on rarefaction curve saturation: 5,000 sequences per sample were sufficient to reach plateau conditions for diversity indices while maximizing the number of samples retained in the analysis.
Reviewer 2 Report
Comments and Suggestions for Authors
In this article, Fedor et al. analyzed fecal and tumor samples from 63 patients with inoperable NSCLC. They found Ruminococcus gauvreauii and Fusicatenibacter were enriched in patients with favorable outcomes, while Coprobacter and the Eubacterium hallii group were associated with disease progression. Before considering publish in our journal, some revision must be made.
- From the Abstract, we cannot well get the keypoint of the result. Please add some results with detailed statistical value in Abstract.
- Please describe more about “Sum Index of Binary Abundance” in the Introduction Part, because this term is not so common in cancer study.
- Please define what is “Inoperable” NSCLC?
- Do the patients use antibiotics in the treatment? The authors need take this information into consideration.
- In Table 4, the “Abundance threshold” are very different with each other, why the author not use the same threshold?
- In the Method part, the author say “Kaplan-Meier log-rank test was carried out with lifelines python package”, but I am not see the result of Kaplan-Meier log-rank test in other part, can the author added Kaplan-Meier survival curves?
Author Response
1. From the Abstract, we cannot well get the keypoint of the result. Please add some results with detailed statistical value in Abstract.
Response 1: In response with this comment, we have updated the abstract:
Higher microbial α-diversity was linked to improved response to ICIs (p-value = 0.0078 for the Chao1 index). Multiple specific taxa, such as Ruminococcus gauvreauii (p-value = 2*10-4), Ruminiclostridium 9 (p-value = 8*10-4), [Eubacterium] ventriosum (p-value = 9*10-4) were enriched in patients with favorable outcomes, whereas Oscillibacter and the Eubacterium hallii group were associated with disease progression (p-value = 2*10-3 and 9*10-3, respectively). The SIBA index, which reflects the absence of multiple beneficial bacterial taxa, proved to be a stronger predictor of treatment response than individual taxa alone. Median SIBA values were 18 vs. 24 in patients benefiting from IO therapy compared to non-responders (p-value = 9*10-7).
2. Please describe more about “Sum Index of Binary Abundance” in the Introduction Part, because this term is not so common in cancer study.
Response 2: In line with the suggestion, we have added the following description of similar composite microbiome indices to the Introduction section (lines 70–73):
After identifying single-taxon correlations, next efforts were concentrated on the role of microbial profiles – specifically, the influence of combinations of bacterial species with both positive and negative effects not only on immunotherapy efficacy but also on the overall health of the meta-organism[8,9].
3. Please define what is “Inoperable” NSCLC?
Response 3: We rephrased the sentence in the following manner (lines 87–88):
This study included 63 patients with inoperable stages of NSCLC – IIIA, IIIB, IV amendable for systemic treatment, who received immune oncology (IO) therapy between 2019 to 2021.
4. Do the patients use antibiotics in the treatment? The authors need take this information into consideration.
Response 4: We added discussion on antibiotics use by patients before the treatment and the fact that they were not used in the treatment in the material and methods section (lines 273–274 and 290–291, respectively):
For example, we were unable to collect data and analyze the role of prior antibiotic use before immunotherapy.
No antibiotics were used in the treatment.
4. In Table 4, the “Abundance threshold” are very different with each other, why the author not use the same threshold?
Response 5: We added the explanation of the threshold values chosen in the manuscript (lines 174–182):
Abundance thresholds were independently determined for each bacterial species based on values yielding the lowest p-value in Fisher's exact test. Thresholds were higher for consistently abundant bacteria (e.g., Faecalibacterium) and lower for bacteria frequently undetected in subsets of samples (e.g., Holdemania). We propose that while common bacterial taxa may provide general metabolic pathways (e.g., vitamin biosynthesis) to the host organism, rare taxa could contribute enzymes for digesting or synthesizing specialized compounds. In this analysis, we omitted regression analyses and machine learning approaches as they require validation using an independent sample cohort.
6. In the Method part, the author say “Kaplan-Meier log-rank test was carried out with lifelines python package”, but I am not see the result of Kaplan-Meier log-rank test in other part, can the author added Kaplan-Meier survival curves?
Response 6: We added two Kaplan-Meier survival curves for SIBA index in the Supplemental Data (Supplemental Figures S1 and S2). We also added references to these figures in the Materials and Methods section (lines 329–331):
As examples, Kaplan-Meier survival curves for patients with high and low SIBA values are shown in Supplemental Figures S1 and S2.
Reviewer 3 Report
Comments and Suggestions for Authors
This manuscript presents a well-conducted and timely study exploring the association between gut microbiome composition and the efficacy of immune checkpoint inhibitors (ICIs) in patients with inoperable non-small cell lung cancer (NSCLC). The authors provide a comprehensive analysis using 16S rRNA sequencing data from a substantial cohort and introduce a novel composite index (SIBA) to integrate microbial taxa associated with clinical benefit. The manuscript is clearly written, methodologically sound, and supported by relevant statistical analyses.
- DNA extraction protocol lacks specific lysis method (e.g., bead beating). Mentioning this is essential as it influences microbiome profiles.
- PD-L1 expression >50% is shown to have p = 0.007 in Table 2, yet the abstract and discussion state it has a “borderline” significance (p = 0.06). This discrepancy shouldbe corrected.
- The triangle matrix of p-values in Figure 1 is difficult to interpret. Consider replacing it with a heatmap showing the direction and strength of associations (e.g., log2 fold change with significance stars).
- It is acknowledged that Akkermansia muciniphila and Bifidobacterium were not detected as significant. Please discuss potential reasons (e.g., dietary differences, detection limits, primer bias).
- Line 86: Replace “disease progression” with “progressive disease” for medical consistency.
- Line 165: “Genus Barnesiellaceae uncultured” is grammatically confusing. Consider rewording to “Uncultured genus from Barnesiellaceae family.”
- Line 173: “a worse response to IO therapy or a higher probability of progression” contradicts the definition of SIBA earlier. Clarify whether higher SIBA values mean better or worse prognosis.
Author Response
1. DNA extraction protocol lacks specific lysis method (e.g., bead beating). Mentioning this is essential as it influences microbiome profiles.
Response 1: We thank Reviewer 3 for this suggestion. We added more details in the DNA extraction protocol (line 299–302):
Total DNA was isolated from fecal specimens following bead-beating homogenization in lysis buffer (0.1 M Tris-HCl, 0.1 M EDTA, 0.01 M NaCl, 0.5% sodium dodecyl sulfate, pH 8.0) with subsequent DNA extraction using the DNeasy Blood & Tissue Kit (Qiagen), following the manufacturer’s protocol.
2. PD-L1 expression >50% is shown to have p = 0.007 in Table 2, yet the abstract and discussion state it has a “borderline” significance (p = 0.06). This discrepancy should be corrected.
Response 2: Corrected (lines 107–110):
However, PD-L1 expression in more than 50% of tumor cells and gender showed a statistically significant difference between cohorts with and without response to immune oncology therapy (p-value = 0.007 and 0.018, respectively).
3. The triangle matrix of p-values in Figure 1 is difficult to interpret. Consider replacing it with a heatmap showing the direction and strength of associations (e.g., log2 fold change with significance stars).
Response 3: In line with the suggestion, we have added more details in the figure caption (lines 157–159). We could not add log2 fold change with significance stars because these designations are not applicable to the analysis carried out.
Figure 1. Association of clinical characteristics with abundance of genus and phylum taxa. A higher number of distinct taxa were significantly associated with benefit from IO therapy than with other characteristics.
4. It is acknowledged that Akkermansia muciniphila and Bifidobacterium were not detected as significant. Please discuss potential reasons (e.g., dietary differences, detection limits, primer bias).
Response 4: In accordance with Reviewer 3, we amended the discussion session (lines 226–228):
The probable explanation might be their lower concentration due to age related Bifidobacteria concentration, Bacteroides dominance, and regional pattern as all the patients originated from North-Western region of Russia [17,18].
5. Line 86: Replace “disease progression” with “progressive disease” for medical consistency.
Response 5: Changed in accordance (lines 94–95).
6. Line 165: “Genus Barnesiellaceae uncultured” is grammatically confusing. Consider rewording to “Uncultured genus from Barnesiellaceae family.”
Response 6: Changed in accordance (Table 4).
7. Line 173: “a worse response to IO therapy or a higher probability of progression” contradicts the definition of SIBA earlier. Clarify whether higher SIBA values mean better or worse prognosis.
Response 7: In agreement with the suggestion, we have replaced the confusing sentence (lines 191–192):
Therefore, a lower SIBA value indicates a group with a higher probability of response to IO therapy.
Round 2
Reviewer 1 Report
Comments and Suggestions for Authors
I would like to express my appreciation to the authors for their thoughtful and constructive responses to the review comments. The added clarifications and expanded discussions significantly enhance the manuscript and will be beneficial for readers.
The inclusion of a discussion regarding the potential impact of PD-L1 expression and sex on microbiome-outcome associations provides valuable context for understanding the findings.
The more detailed description of the SIBA index methodology, including the rationale behind taxon selection and threshold determination, contributes to the transparency of the approach. Furthermore, the acknowledgment of limitations related to internal validation and multiple-comparison testing offers important guidance for readers in interpreting the statistical results.
The enriched discussion section deserves special recognition. The integration of relevant literature specific to NSCLC and the comparison with previously reported taxa effectively situates the findings within the wider context of the field, underscoring the significance of this work.
Lastly, the thorough explanation of the choice to use a 5000-read rarefaction threshold, supported by considerations of rarefaction curve saturation and sample retention, enhances the methodological clarity of the study.
In summary, these revisions reflect a commendable commitment to improving the clarity, rigor, and interpretive depth of the manuscript, contributing to a high-quality addition to the literature. I express my gratitude to the authors for their important contributions to the field.
Reviewer 2 Report
Comments and Suggestions for Authors
The authors have well replied my comments, now It can be accept for publication.